# Inflammatory Mesenchymal Stem Cells Express Abundant Membrane-Bound and Soluble Forms of C-Type Lectin-like CD248

**DOI:** 10.3390/ijms24119546

**Published:** 2023-05-31

**Authors:** Melissa Payet, Franck Ah-Pine, Xavier Guillot, Philippe Gasque

**Affiliations:** 1Unité de Recherche en Pharmaco-Immunologie (EPI), Université et CHU de La Réunion, 97400 Saint-Denis, France; 2Service d’Anatomie et Cytologie Pathologiques, CHU de La Réunion, 97410 Saint-Pierre, France; 3Service de Rhumatologie, CHU de La Réunion, 97400 Saint-Denis, France; 4Laboratoire d’Immunologie Clinique et Expérimentale Océan Indien (LICE-OI), CHU de la Réunion, 97400 Saint-Denis, France

**Keywords:** mesenchymal stem cells, CD248, inflammation, rheumatoid arthritis

## Abstract

CD248 (endosialin) belongs to a glycoprotein family that also includes thrombomodulin (CD141), CLEC14A, and CD93 (AA4) stem cell markers. We analyzed the regulated expression of CD248 in vitro using skin (HFFF) and synovial (FLS) mesenchymal stem cell lines, and in fluid and tissue samples of rheumatoid arthritis (RA) and osteoarthritis (OA) patients. Cells were incubated with either rhVEGF_165_, bFGF, TGF-β1, IL1-β, TNF-α, TGFβ1, IFN-γ, or PMA (Phorbol ester). There was no statistically significant change in membrane expression. A soluble (s) form of cleaved CD248 (sCD248) was detected after cell treatment with IL1-β and PMA. Matrix metalloprotease (MMP) MMP-1 and MMP-3 mRNAs were significantly up-regulated by IL1-β and PMA. A broad MMP inhibitor blocked the release of soluble CD248. In RA synovial tissue, we identified CD90^+^ perivascular MSCs double-stained for CD248 and VEGF. High sCD248 levels were detected in synovial fluid from RA. In culture, subpopulations of CD90^+^ CD14^−^ RA MSCs were either identified as CD248^+^ or CD141^+^ cells but CD93^−^. CD248 is abundantly expressed by inflammatory MSCs and shed in an MMP-dependent manner in response to cytokines and pro-angiogenic growth factors. Both membrane-bound and soluble CD248 (acting as a decoy receptor) may contribute to RA pathogenesis.

## 1. Introduction

CD248, also known as endosialin or tumor endothelial marker 1 (TEM1), is a 165 kDa type 1 transmembrane glycoprotein. It belongs to a protein family involved in cell–cell interactions that also includes CD93 [1,2,3,4], CLEC14A, and thrombomodulin (CD141) [5,6,7]. The four family members share a strong homology in the N-terminal region, wherein lies a C-type lectin-like domain (CTLD). Indeed, the N-terminal region coding sequence of CD248 shares 38% and 33% identity with CD141 and CD93, respectively. Interestingly, the N-terminal region of CD141 has been functionally implicated in mediating constitutive endocytosis and in modulating inflammation [5,6]. The CTLD, along with the other constituent domains of CD248, namely the epidermal growth factor-like (EGF-like) and complement control protein (CCP) domains, have each been shown to be involved in cell–cell interactions and cell adhesion events in other proteins [8,9,10,11,12].

Expression of human CD248 has been shown to be highly restricted close to tumor vessels, implicating CD248 as being a modulator of tumor angiogenesis [13]. CD248 was discovered by Rettig et al. while exploring endothelial cell markers which could act as drug targets [14]. A monoclonal antibody, mAb FB5, was used in immunohistochemical studies and specifically stained the endothelium in a substantial proportion of sarcomas, carcinomas, and neuroectodermal tissues but did not stain endothelial cells in any normal tissues [14]. In addition, mAb FB5 did not stain any other non-endothelial cells in normal tissues but did stain malignant cells in a subset of sarcomas, and reactive stromal fibroblasts in a subset of epithelial cancers [14]. The specificity of CD248 expression for tumor endothelium in vivo was confirmed by an independent study by St Croix et al., who used serial analysis of gene expression (SAGE) technology to identify CD248 as being specifically upregulated in the endothelium of colon, lung, liver, and brain tumors [15]. However, CD248 was also expressed in the endothelium of granulation tissue in healing wounds and in the corpus luteum [15], arguing for a potential role of CD248 in fibrosis [16,17,18]. In a recent analysis of brain tumor biopsies, we found a strong expression of CD248 in highly invasive glioblastoma multiforme, high-grade astrocytomas, and metastatic carcinomas [19]. We also observed that CD248 was expressed by endothelial cells, CD90^+^ fibroblasts close to the meninges, and α-smooth muscle actin-positive cells in some vessels [19].

Rheumatoid arthritis (RA) is characterized by chronic infiltration of the joint by innate and adaptive immune cells, hyperplasia of synovial mesenchymal stem cells (MSCs), and macrophage activation leading to synovial pannus formation, with aggressive invasion and destruction of the surrounding cartilage and neoangiogenesis to sustain cellular growth and systemic inflammation [20]. The inflammatory MSCs contribute to the pathology of RA by releasing soluble mediators, including cytokines and proteolytic enzymes (including matrix metalloproteases (MMPs), cathepsins, and serine proteases), which directly or indirectly promote cartilage degradation [21,22]. By contrast, osteoarthritis (OA) is considered a non-inflammatory form of arthritis and the role of some inflammatory mediators in OA pathogenesis has been emphasized [23].

The aim of our study was to investigate the role of CD248 expression by MSCs in the pathology of proliferative and invasive remodeling of synovial tissue in arthritis. We found that CD248 is abundantly expressed by two models of MSCs and cleaved in an MMP-dependent manner in response to inflammatory cytokines and pro-angiogenic growth factors. Both membrane-bound and soluble CD248 (decoy receptor) may contribute to RA pathogenesis.

## 2. Results

### 2.1. CD248 Is a Membrane Protein Expressed by Skin-Derived MSCs (HFFFs), Neuroblastoma, and Other Tumor Cell Lines

CD248 mRNA expression was examined in a variety of human cell lines and primary MSCs by RT-PCR to identify a model that allows the evaluation of CD248 expression in vitro, in the absence of endogenous expression by endothelial cell lines. Significant CD248 mRNA expression was found in IMR-32 neuroblastoma cells and in human fetal foreskin fibroblast (HFFF) cell cultures, a model of skin-derived MSCs (Figure 1). No CD248 expression was detected in unstimulated endothelial cells (EaHy929), other human cell lines (K562, Jurkat), or blood-derived mononucleated cells (MNC, i.e., B and T lymphocytes and monocytes) (Figure 1A). These results were further confirmed by flow cytometry. Strong membrane expression of CD248 was detected in IMR-32, HFFFs, and a subset of the adult fibroblast cell line 8399 (Figure 1B). Interestingly, the breast epithelial tumor cell lines, MDA-231 and MDA-453, also express CD248, but not MCF-7. In addition, a subset of natural killer cells (YT) expresses CD248. Other cell lines, such as 55N (neuroblastoma), Jurkat (T cells), and Akata (B cells), did not express CD248 (Figure 1B).

Consequently, the HFFF cell line was selected to explore regulated CD248 expression.

To better characterize CD248 expression in HFFFs, we evaluated its specific intracellular localization by immunocytochemistry. We investigated the expression of CD248 and specific organelle markers, such as BiP (endoplasmic reticulum), EEA1 (endosomes), Lamp1 (lysosomes), GM130 (Golgi apparatus), and anti-paxillin (focal contacts). In addition to its membrane distribution, CD248 colocalized with EEA1, an endosome-specific antibody, indicating that a pool of CD248 is stored in intracellular endosomal compartments (Figure 1C). No colocalization was observed with BiP, GM130, or paxillin.

### 2.2. A Soluble Form of CD248 Is Released from the Skin-Derived MSCs (HFFFs) following Treatment with IL1-β and PMA

In order to determine the mechanism by which CD248 expression in MSCs is regulated, HFFFs were treated with rhVEGF_165_, bFGF, TGF-β1, IL1-β, and PMA for 24 h. Analysis of cell surface CD248 expression by flow cytometry revealed no significant change compared with non-treated cells (Figure 2) at 24 h, *p* > 0.05 (*n* = 3). Total cell CD248 expression was determined by Western blot analysis of cell lysates. Comparison of band intensities between non-treated and treated cells showed no significant change in total CD248 expression following any of the 24 h treatments. Figure 2B shows a representative blot for CD248 and α-tubulin of treated-cell lysates (24 h time course). 

A soluble form of CD248 (sCD248) was identified by Western blot of the concentrated tissue culture supernatant (TCS) following each treatment. After 4 h of treatment, a band of 110 kDa, smaller than that detected for the membrane-bound form of CD248, was detected. The soluble form of CD248 was strongly upregulated after a 24 h treatment with IL1-β (up to 70-fold) and PMA (up to 62-fold). An up-regulation of soluble CD248 expression was also observed following treatment with rhVEGF_165_, bFGF, and TGF-β1; however, this was less marked than with IL1-β or PMA. Figure 2C shows the Western blot and analysis of band intensities of a representative experiment (*n* = 3). 

### 2.3. Matrix Metalloproteases (MMPs) May Play a Role in the Generation of Soluble CD248

To ascertain the possible role of MMPs in the cleavage and shedding of CD248 ectodomain following cytokine and growth factor stimulation, we incubated the cells for 30 min prior to treatment with the growth factors with broad-spectrum MMP inhibitor 1,10-phenanthroline. TCS was then concentrated and CD248 expression was determined by Western blot. MMP inhibitor exposure significantly reduced the formation of soluble CD248 in response to all treatments but most significantly reduced soluble CD248 formation following bFGF, IL1-β, and PMA treatment (Figure 3).

To characterize the plausible MMPs involved in CD248 ectodomain cleavage, we carried out semi-quantitative RT-PCR using primers specific for five MMPs/MMP inhibitors, namely MMP-1, MMP-2, MMP-3, MMP-9, and TIMP-1. Total RNA was extracted from the cells following treatment for 4 h, as described above, and subjected to RT-PCR. All MMPs and TIMP1 were expressed constitutively. MMP-1 expression was up-regulated by rhVEGF_165_ and TGF-β1, and strongly up-regulated by IL1-β and PMA treatment. Similarly, MMP-3 expression was significantly up-regulated by treatment with IL1-β and less so by PMA. By contrast, there was no change in the expression of MMP-2 or TIMP-1 following any of the treatments, and MMP-9 expression decreased following all treatments (Figure 3B).

### 2.4. Inflammatory Cytokines Failed to Upregulate CD248 mRNA Expression by Synovial-Tissue-Derived MSCs but Upregulated the Levels of sCD248

To further determine the expression and the regulation of CD248 together with that of canonical synovial MSC markers (i.e., CD90 and podoplanin (PDPN)), we used a primary culture of human synovial MSCs. qRT-PCR was performed after stimulation of MSCs with IL1-β, TNF-α, TGF-β1, or IFN-γ (all at 20 ng/mL) for 6, 24, 48, and 72 h. These treatments did not show any cytotoxic effects on the cells as tested by MTT and LDH assays (Appendix A). Only IFN-γ increased CD248 mRNA expression at early and late time points (Figure 4). CD248 mRNA expression failed to be upregulated and was rather decreased at 48 and 72 h by IL1-β, TNF-α, and TGF-β1. By contrast, IL1-β and IFN-γ significantly increased CD90 mRNA expression at 48 h. Pro-inflammatory mediators such as IL1-β and TNF-α significantly increased PDPN mRNA expression at 48 h and 72 h (Figure 4C).

To address the regulated expression of sCD248, we stimulated synovial MSCs from Sciencell for 24 h and the TCS was tested using our in-house ELISA and CD248FL-FC fusion protein as a standard. sCD248 was detectable in control 12 ± 4 pg/mL/10^6^ cells and was significantly increased in all stimulated cell conditions, particularly significantly for IL-1 beta 852 ± 148 pg/mL/10^6^ cells and PMA 781 ± 97 pg/mL/10^6^ cells (Appendix A). 

### 2.5. MMP3-Independent Shedding of CD248 in Synovial-Derived MSCs from RA and OA Patients

Primary RA-derived MSCs were also found to express abundant levels of membrane-bound and soluble forms of CD248. A homogenous population of fibroblasts (CD90^bright^, CD14^neg^, and CD4^neg^) was obtained after 1–2 passages from RA and OA synovial tissues (Figure 5). At passages 3–5, a significant proportion of the CD248-expressing cell population was CD90-positive (73.4% of RA and 75.2% of OA) confirming the cells were MSCs (Figure 5A). We found no differences in the expression of CD248 from synovial cells derived from patients with RA or OA. Western blot analyses of RA fibroblast cell lysates indicated a robust expression of CD248. A soluble form (110 kDa major band) was also detected in TCS in response to IL1-β stimulation, but the MMP-3-specific inhibitor failed to affect the release of CD248 (Figure 5B). Other specific bands with an apparent molecular mass of 90 kDa and 75 kDa were also detected in RA MSC TCS.

Having established the presence of a soluble form of CD248 in vitro, we addressed the presence of a soluble form of CD248 in vivo by immunoprecipitation techniques using RA synovial fluids. The immunoprecipitated proteins were detected by Western blot probing with affinity-purified rabbit anti-CD248. A band of 110 kDa, along with a number of smaller bands arranged in a ladder-like pattern, was visible following immunoprecipitation with P7C5 mouse anti-CD248 antibody (Figure 5C). These bands were not visible by immunoprecipitation with an irrelevant anti-CD93 control antibody (not shown). As a means of comparing soluble CD248 with membrane-bound CD248, lysates were prepared from synovial tissues obtained from the diseased joints of an RA patient. Membrane-bound CD248 expression was confirmed as a broad band of 120–170 kDa (Figure 5C). 

### 2.6. CD248 Is Abundantly Expressed by Inflammatory MSCs in RA Synovial Tissues

To ascertain the identity of the cell types expressing CD248 in the inflamed synovial tissue, we performed double immunostainings of tissue and cultured cells. CD248 expression in situ was first examined in frozen tissue sections from RA patients (Figure 6). Appropriate negative controls using isotype-matched irrelevant antibodies demonstrated no staining (Figure 6a,b insets). At low magnification, strong CD248 staining was observed on stromal MSCs and next to a subset of large and small vessels double-stained for the CD93 endothelial marker (Figure 6a,d, white arrow). CD248 expression was mainly associated with CD90 and prolyl 4-hydroxylase positive MSCs, while no staining was detected on infiltrating CD14^+^ and CD4^+^ cells. Interestingly, CD248^+^ MSCs were in close proximity to clusters of VEGF-positive cells (Figure 6j,k). Endothelial cells identified by the CD93 and CD31 stainings remained largely CD248 neg (Figure 6g–i,m–o). CD248 was also detected on some α-smooth muscle actin-positive pericytes present in the vasculature. 

In culture, subpopulations of CD90^+^ CD14^−^ RA synovial cells were either CD248^+^ or CD141^+^ cells but failed to stain for CD93 (Figure 6p). Occasionally, less than 5% of the cultured cells were double-stained for CD248 and CD141. CD248 was distributed throughout the cytoplasm with prominent cell surface staining. On close examination (Figure 6r, ×1000), CD248 intracellular staining colocalized with EEA1 (as depicted in Figure 1c for skin-derived MSCs) but not with CD63, a lysosomal membrane glycoprotein (Figure 6q).

We recently took advantage of a new commercially available rabbit monoclonal anti-CD248 antibody to stain the fixed paraffin wax tissue sections of an RA patient. Sections were double-stained for either CD45 to identify infiltrating leukocytes or ERG, an endothelial nuclear cell marker. The inflammatory status of the synovial tissue was confirmed by the detection of numerous CD45^+^ infiltrating leukocytes. CD248 was detected on perivascular MSCs but was not expressed on ERG+ endothelial cells (Figure 7).

## 3. Discussion

The high expression of CD248 by vessels in a variety of tumor types has been previously reported by several groups [14,15,19]. Our aim in this study was to explore the expression of CD248 in other proliferative and inflammatory pathologies, such as RA. Using validated antibodies generated in-house (and new commercially available rabbit monoclonal antibody, Figure 7), we confirmed the work of Rettig et al. and MacFayden et al., for instance, by showing that CD248 is expressed in fetal and adult MSCs in vitro at the molecular and protein levels, and also that CD248 is not expressed by endothelial cell lines in vitro [14,24]. 

Closer inspection of the cellular localization of CD248 by double immunofluorescence staining revealed that as well as being membrane-bound, intracellular stores exist in the endosomal compartment. This is in contrast to the work of Christian et al., who found by indirect immunofluorescence that CD248 expression was restricted to the cell surface [25]. However, our findings are in agreement with those of Opavsky relating to the murine homolog of CD248, which was localized in the cytosol in a granular fashion, as well as at the cell membrane in transfected cells [26]. The nature of the intracellular granules expressing murine CD248 was not characterized further. However, our data now suggest that they may be endosomal. Notably, the CD248 extracellular domain contains a WIGL amino acid sequence, which is conserved between the four members of this protein family and is present in other cell surface proteins known to modulate endocytosis [25,27]. 

We speculate that CD248–receptor–ligand interaction triggers a signal within the MSCs causing internalization by endocytosis. Moreover, CD248 has been shown to interact with several proteins of the ECM, including the ligand multimerin-2, a unique endothelial-specific ECM protein [28,29]. CD93 has also been shown to interact with multimerin-2, suggesting that all three proteins may be important in cell–cell interactions to maintain close contact between endothelial cells and MSCs at the perivascular level (see summary Figure 8). We do not know what could be the role of the CD93–multimerin–CD248 complex and downstream signaling in contributing to the capacity of CD248 to boost PDGF-induced proliferation of MSCs, as shown in the model of hepatic stellate MSCs [30]. 

In an effort to determine the inflammatory effectors which modulate CD248 expression, we treated two different models of MSCs (skin and synovial tissue) in vitro with a range of growth factors and cytokines that have been identified as perpetrators of the pathology of cancer and arthritis. While there was no significant variation (with the exception of cells treated with IFN-gamma, but only for 72 h) in either mRNA level (RT-PCR), total cellular (WBlot), or cell surface (FACS) expression of CD248, a significant and rapid up-regulation in the release of a soluble form of CD248 was observed in response to the growth factors rhVEGF_165_, bFGF, and TGF-β1. The most significant up-regulation was in response to the pro-inflammatory cytokine IL1-β and the non-specific modulator of protein kinase activity PMA. IL1-β is a well-established cytokine in the pathology of RA, which, upon release from activated synovial MSCs and macrophages, stimulates further inflammatory responses and the release of proteolytic enzymes, which in turn degrade the cartilage and release cell surface proteins [31]. PMA has been found to directly cause the release of soluble proteins in HUVEC. However, it is a pleiotropic molecule with multiple actions making it difficult to establish the mechanism [32]. In vivo, it is reasonable to speculate that raised levels of IL1-β in a diseased RA synovium may contribute to the generation of soluble CD248 in synovial fluid, the presence of which we have identified here by immunoprecipitation. The apparent lower molecular mass of sCD248 (110 kDa) is indicative of single or multiple cleavages of the transmembrane form (170 kDa), possibly involving MMPs, known to be abundantly expressed in RA inflammatory conditions.

Next, we wanted to establish the mechanism by which the growth factors and cytokines elicit the formation of soluble CD248. The activity of the MMPs in eliciting membrane cleavage of CD248 was confirmed by the addition of 1,10-phenanthroline, a broad-spectrum MMP inhibitor. Interestingly, the formation of soluble CD248 was dramatically reduced, and IL1-β failed to stimulate an up-regulation of expression. Moreover, RT-PCR analysis showed a transcriptional up-regulation in MMP-1 and MMP-3 in response to IL1-β and, to a lesser extent, PMA. MMP-1 (collagenase-1) and MMP-3 (stromelysin-1) are just two of a plethora of proteases released in vivo from activated synovial MSCs in response to inflammatory cytokines such as IL1β, where they contribute to cellular remodeling and promote the destruction of cartilage within the joint [33,34]. However, a specific MMP-3 inhibitor failed to control CD248 shedding from RA MSCs stimulated with IL1β and hence, we cannot rule out the possibility that other enzymes, e.g., cathepsins and serine proteases, all of which play a role in RA, may be involved in the generation of sCD248 [35]. The release of the extracellular domain of cell surface proteins in an MMP-dependent manner is a common phenomenon termed ectodomain shedding. Its purpose varies from increasing the bioavailability of a protein normally bound to the cell surface—for example, growth factors—to regulating ligand–receptor interactions at the cell surface [32]. Ectodomain shedding of a soluble form containing the lectin domain has also been described for CD93 and CD141, implying that this is an important cellular process [36,37]. Whether or not sCD248 retains ligand binding activity and function has not been determined. 

We have recently observed that CD248 CTLD is involved in homotypic interactions which may be of functional importance (unpublished observations). However, the possibility that CD248 engages in heterotypic interactions with an as-yet elusive CD248 ligand (different from multimerin-2) cannot be excluded and merits further study. Ultimately, receptor shedding is important in modulating an array of cellular responses, although the exact nature of the response in CD248-expressing cells remains unclear. Soluble CD248 formation may be a means of reducing cell–cell interactions, for instance between endothelial cells and perivascular MSCs, promoting cell proliferation, or it may act as a sink for the circulating ligand (yet to be identified) (See summary Figure 8). 

The loss of the membrane form of CD248 by the perivascular synovial MSCs may confer invasive and aggressive functions on these cells in RA. Interestingly, a recently identified population of synovial MSCs, CD90^−^ CD248^−^ but PPDN+, are localized in the lining layers of the synovial tissue and contribute to cartilage degradation. By contrast, the CD90^+^ CD248^+^ but PDPN^−^ were localized in the sublining layer and surrounding endothelial cells [22,38]. Therefore, inflammatory mediators such as IL1-β may induce a phenotype switch of perivascular MSCs to a more aggressive phenotype by decreasing membrane CD248 and increasing PDPN expression. Our data support this hypothesis, but this is an over-simplistic paradigm that needs further investigation.

Maia et al. demonstrated that CD248 knockout mice had less severe arthritis compared to wild-type [39]. CD248 knockout mice had lower levels of pro-inflammatory cytokines in plasma, reduced accumulation of inflammatory cells, and less articular and bone damage compared to wild-type mice. In humans, studies have confirmed that CD248 and PPDN (together with FAP and DAF/CD55) are highly expressed in the synovial tissue in the early stage of arthritis [40]. The expression of these MSC markers was not associated with prognostic outcomes of disease persistence or resolution. These results confirmed however that CD248 is an important component of the inflammatory reaction in arthritis, albeit through still poorly characterized mechanisms. The detection of the soluble form of CD248 adds to the complexity of deciphering the exact role of CD248 on MSCs in inflammatory settings. 

## 4. Materials and Methods

### 4.1. Generation of Antibodies against Human CD248 and ELISA Protocol

In order to generate antibodies against human CD248, two recombinant forms of the protein were expressed. First, a soluble recombinant form of CD248 (CD248-Fc) was generated comprising the N-terminal region (CTLD, CCP, and first EGF domain) fused to a human IgG1 Fc tail. Second, human CD248-FL was cloned into pDR2ΔEF1α and expressed in mouse myoblast C2C12 cells, essentially as described [19]. 

Monoclonal antibodies were generated by repeated subcutaneous immunization of Balb/c mice with either 100 μg CD248-Fc (human IgG1 Fc fusion protein fused to CD248 CTLD) or mouse C2C12 myoblast cells expressing full-length CD248 (10^7^/500 μL 0.9% NaCl) as described [41]. Following the final immunization, splenocytes were isolated and fused with Sp2/O myeloma cells to generate hybridomas by standard methods [41]. Hybridomas were screened for immunoreactivity against CD248 by flow cytometric staining of K562 cells expressing CD248, and K562 expressing the human C3aR as a negative control. Specificity was confirmed by ELISA, where the plate was coated with CD248-Fc and an HRPO-conjugated goat anti-mouse antibody was used to detect specific binding [19]. Positive hybridomas were cloned by limiting dilution, and screening and cloning were continued until monoclonal antibodies were obtained. The monoclonal antibody isotype was determined using the “Isostrip” isotyping kit (Boehringher Mannheim, Lewes, UK). 

Monoclonal antibodies (both IgG1) against CD248 (CTLD)-Fc and CD248-FL termed P7C5 and P1A6, respectively, were generated. The specificity of these antibodies for CD248 was confirmed by immunohistochemical staining of CHO cells transfected with CD248-FL or the human C3aR as a negative control. Both antibodies strongly stained the CD248-transfected cells but did not stain the C3aR cells.

A rabbit antibody against human recombinant CD248 was generated and affinity-purified as described previously [19]. 

We also measured the level of soluble CD248 by ELISA.

To this end, the ELISA plate was coated with 2 μg/mL of rabbit anti-endosialin in coating buffer (48% 0.2 M NaHCO_3_, 27% 0.2 M Na_2_CO_3_, 25% dH_2_O, and pH adjusted to 9.6) by overnight incubation at 4 °C. Excess antibody was washed off with PBS–0.1% Tween 20 and the non-specific sites were blocked by incubation in PBS–5% milk for 2 h at room temperature. A standard curve was prepared by diluting recombinant CD248-Fc in PBS–5% milk at concentrations from 0 to 100 ng/mL. Tissue supernatants were added neat. Samples and standards were added in triplicate and incubated for 2 h at room temperature. The plate was washed well in PBS–0.1% Tween 20. An amount of 2 μg/mL of the monoclonal anti-CD248 antibody (clone P7C5) in PBS–5% milk was incubated for 2 h at room temperature and following a further round of washes in PBS–0.1% Tween 20, an HRPO-conjugated donkey anti-mouse secondary antibody (1:1000, Jackson Immunoresearch Laboratory) was incubated for 2 h at room temperature. Color development was initiated using the TMB (Peprotech, Cranbury, NJ, USA) and was monitored spectrophotometrically. sCD248 was quantified from the standard curve and expressed as pg/mL relative to 10^6^ cells per well.

### 4.2. Cell Culture and Tissue Specimens

CHO, MCF-7, MDA-231, and MDA453 were maintained in Hams F-12 medium. C2C12, 8399, EaHy926, IMR-32, and 55N were cultured in D-MEM. A549, K562, Jurkat, Akata, and YT were cultured in RPMI-1640 medium. HFFF cultures were kindly obtained from Professor G. Wilkinson (Cardiff University, Cardiff, UK) and prepared as described [42]. All media was supplemented with 10% heat-inactivated fetal bovine serum (FBS), 50 U each of penicillin/streptomycin, 0.5 μg/mL amphotericin B, 2 mM L-glutamine, and 1 mM sodium pyruvate (all cell culture reagents from Invitrogen, Paisley, UK). Cells were maintained at 37 °C with 5% CO_2_.

Fibroblast-like synoviocytes (FLS) were either obtained from ScienCell (ScienCell, Carlsbad, CA, USA, 4700; Cliniscience) or isolated from synovial tissue collected from the diseased joints of OA and RA patients during routine surgical interventions and digested with 1 mg/mL collagenase. Cells were kindly provided by Dr. Anwen S. Williams. Ethical approval was granted by the Bro Taf Health Authority (LREC 02/4692) on the grounds that this study was initially carried out at the Brain Inflammation and Immunity Group (headed by Prof. P. Gasque at Cardiff University). Non-adherent cells were removed, and adherent cells were harvested with trypsin–EDTA and passaged until a homogenous population of synovial fibroblasts remained, as determined by flow cytometric staining of cells for CD90 (Thy-1) (Serotec, Oxford, UK), CD4, and CD14 (monoclonal antibodies kindly provided by Dr. V. Horejsi, Prague, Czech Republic). All experiments were conducted with synovial fibroblasts (CD4^neg^, CD14^neg^, and CD90^bright^) at passages 3–5. Cell stimulations were performed on growth-arrested fibroblast cultures, as described [43].

Fresh synovial tissue for immunohistochemistry was snap frozen in isopentane which was kept on dry ice. Then, 10 μm sections were cut, collected onto poly-L-lysine-treated glass slides (Surgipath, Peterborough, UK), and allowed to dry at room temperature before staining.

### 4.3. CD248 Expression in Cell Lines and HFFFs by Semiquantitative RT-PCR

Total RNA was extracted from EaHy926, A549, K562, IMR-32, Jurkat, MNCs (mononuclear cells), and HFFFs using the Ultraspec™ RNA isolation system according to the manufacturer’s instructions (BIOTECX laboratories, Houston, TX, USA). cDNA was generated using 2 μg RNA and a previously described protocol [1]. An amount of 5 μL of cDNA was used in a 50 μL PCR containing 0.2 μM of each CD248 specific primer (P6 5′ GTCGGATCCATGCGGCCCCAGCAGCTGCTAC 3′ and P7 5′ GGCCTATAGGCCTCGCAACTGCGCCCGTCTGC 3′), 0.2 μM dNTPs, and 2.5 units of Platinum^®^ Taq DNA polymerase (Invitrogen). Thermal cycling was at 94 °C for 4 min, followed by 94 °C for 45 s, 60 °C for 1 min, and 72 °C for 2 min for 5 cycles; 94 °C for 30 s, 60 °C for 30 s, and 72 °C for 1 min 30 s for 25 cycles; and a final elongation step at 72 °C for 15 min. RT-PCR products were separated on a 0.7% ethidium-bromide-stained agarose gel and visualized under UV light. 

### 4.4. Expression of CD248 by HFFFs as Tested by Flow Cytometry Analyses

HFFFs were washed twice in 0.9% NaCl and adherent cells detached using PBS containing 1 mM EDTA. Then, 10^5^ cells resuspended in FACS buffer (PBS, 1% BSA, and 0.1% NaN_3_) were incubated with 3.5 μg of affinity-purified rabbit anti-CD248 for 1 h at 4 °C. Cells were washed three times in FACS buffer and then incubated with an rPE-conjugated goat anti-rabbit secondary antibody (Sigma, St. Louis, MO, USA). Cells were washed three times in FACS buffer, resuspended in PBS–1% BSA, and analyzed on a FACSCalibur instrument (BD Biosciences, Oxford, UK). CD248 expression by synovial fibroblasts was confirmed by double staining with affinity-purified rabbit anti-CD248, as described above, and 10 μg/mL of mouse anti-human CD90 (Thy-1) followed by rPE-conjugated goat anti-rabbit and FITC-conjugated goat anti-mouse secondary antibodies (1:200, The Binding Site, Birmingham, UK).

### 4.5. Subcellular Localization of CD248 by Immunocytochemistry

HFFFs (passage 12–20) or RA fibroblasts were grown on 16 mm sterile glass coverslips, washed five times in PBS, and fixed in ice-cold acetone for 10 s. Cells were washed briefly in PBS and non-specific interactions were blocked by incubation for 10 min with PBS–1% BSA. Cells were incubated with 7 μg/mL of affinity-purified rabbit anti-CD248 and a series of antibodies against cell organelle markers, including 2.5 μg/mL each of anti-BiP (endoplasmic reticulum), anti-EEA1 (endosomes), anti-Lamp1 (lysosomes), anti-GM130 (Golgi apparatus), and anti-paxillin (focal contacts) (BD organelle sampler kit) in PBS–1% BSA for 1 h at room temperature. After several washes in PBS–1% BSA, cells were incubated for 1 h at room temperature in 5 μg/mL each of TRITC-conjugated donkey anti-rabbit and FITC-conjugated donkey anti-mouse antibodies (Jackson Immunoresearch Laboratories, Luton, UK). After further washes in PBS and a final wash in dH_2_O, coverslips were mounted with Vectashield (Vector Laboratories, Peterborough, UK). Staining was visualized under a Leica TCS SP2 confocal microscope or DMLB microscope (Leica, Wetzlar, Germany). 

### 4.6. Regulation of CD248 Expression by Stimulation of HFFFs by Growth Factors and Cytokines

HFFFs (passages 12–20) or human RA synovial fibroblasts (passages 3–5) were rendered quiescent by incubation in a serum-free medium for 24 h [43]. The cells were then treated with 20 ng/mL rhVEGF_165_, 10 ng/mL bFGF, 1 ng/mL TGF-β1 (all R&D Systems, Abingdon, UK), 200 U/mL IL1-β (kind gift of Hoffmann La Roche, Nutley, NJ, USA), and 50 ng/mL PMA (Sigma) in serum-free medium or were incubated in serum-free medium alone for 4 h and 24 h. In experiments to test the effect of MMP inhibition, cells were incubated with 5 μM of either the broad-matrix metalloprotease inhibitor 1,10-phenanthroline (Sigma) or a specific MMP-3 inhibitor (3-[4-(4-cyanophenyl] phenoxy] propanohydroxamic acid) (Calbiochem, Nottingham, UK) in serum-free medium, for 30 min at 37 °C prior to treatment together with growth factors. 

CD248 cell surface expression was determined by flow cytometry using 3.5 μg/mL of affinity-purified rabbit anti-CD248. Changes in expression with each treatment were expressed as fold changes in the mean of fluorescence over control (non-treated) cells. Total cell lysates were prepared by washing twice in 0.9% NaCl, detaching cells from the flask with a cell scraper, and incubating in Triton X-100 lysis buffer (1% Triton X-100, 10 mM EDTA, 1 mM PMSF, 10 μg/mL pepstatin, and 10 μg/mL leupeptin in PBS) for 1 h at room temperature followed by centrifugation at 13,000 rpm at 4 °C for 15 min. TCS was dialyzed in 2 mM EDTA overnight at 4 °C, freeze-dried, and resuspended in non-reducing Laemmli buffer. To examine CD248 expression in total cell lysates and TCS following treatment, proteins in whole-cell lysates and concentrated TCS were resolved by SDS-PAGE on a 7.5% gel under non-reducing conditions and subjected to Western blot, essentially as described previously [1]. The blots were probed with 0.7 μg/mL of affinity-purified rabbit anti-CD248 or with mouse anti-α-tubulin (clone B-5-1-2, 1:1000, Sigma). The intensity of each band was determined using the GS-710 calibrated imaging densitometer (Bio-Rad, Hemel Hempstead, UK). Differences in protein loading were accounted for by the intensity of bands on the α-tubulin blot (housekeeping protein). Changes in CD248 expression were expressed as fold changes in band intensity over control (non-treated) cells.

### 4.7. Matrix Metalloprotease Expression by HFFFs

Following treatment of HFFFs with growth factors and cytokines as described above, total RNA was extracted from the cells and cDNA was generated, also as described above. A 5 μL aliquot of cDNA was amplified by PCR using specific sets of primers for MMP1 (5′ ATTCTACTGATATCGGGGCTTTGA 3′ (forward), 5′ ATGTCCTTGGGGTATCCGTGTAG 3′ (reverse)), MMP2 (5′ TGGGGCCTCTCCTGACATTGAC 3′ (forward), 5′ TCCACGACGGCATCCAGGTTA 3′ (reverse)), MMP3 (5′ ATGCCCACTTTGATGATGATGAAC 3′ (forward), 5′ CCACGCCTGAAGGAAGAGATG 3′ (reverse)), MMP9 (5′ GCGCTGGGCTTAGATCATTCCTCA 3′ (forward), 5′ GCAGCGCGGGCCACTTGTC 3′ (reverse)), and TIMP1 (5′ CGTCATCAGGGCCAAGTTCGTG 3′ (forward), 5′ GAGGCAGGCAGGCAAGGTGAC 3′ (reverse)). Following an initial denaturation step at 94 °C for 4 min, there were 5 cycles at 94 °C for 45 s, at the annealing temperatures given below for 1 min, and elongation at 72 °C for 2 min, followed by a total of 25 cycles with a 30 s denaturation and annealing at 52 °C and 54 °C (MMP1 and MMP3), 56 °C and 58 °C (MMP2 and TIMP1), or 60 °C (MMP9) for 1 min and 72 °C for 2 min. A final elongation step at 72 °C for 15 min followed. Actin-specific primers were used to determine the quantity of mRNA. PCR products were separated on a 0.7% ethidium-bromide-stained agarose gel and visualized under UV light.

### 4.8. Quantitative RT-PCR Analyses of CD248 mRNA Expression by Synovial MSCs

Total RNA from synovial MSCs (Sciencell, #4700), control or stimulated with IL1-β, TNF-α, TGFβ1, or IFN-γ at 20 ng/mL was extracted using a Quick-RNATM viral kit (Zymo research, Cat No R1035). The potential cytotoxic activities of all the different treatments were tested by two independent assays (MTT and LDH) as previously described [44]. qRT-PCR experiments were performed using the SensiFast Probe No-ROX One-Step Kit (Meridian Bioscience, Cat. No BIO-76005) to which Sybergreen has been added beforehand. qRT-PCR was achieved in a final volume of 5 µL containing 2.7 µL of enzyme mix, 1.3 µL of primer mix with a final concentration of 250 nM, and 1 µL of RNA. qRT-PCR was performed in a Quantstudio 5 PCR thermocycler (Thermo Fisher Scientific, Waltham, MA, USA), as described [44]. GAPDH was used as the housekeeping gene. Samples were analyzed in three independent experiments. The sequences of the used primers are as follows: GAPDH (5′ CCATGCGGAAGGTGAAGGTC 3′ (forward), 5′ ACATGTAAACCATGTAGTTGAGGT 3′ (reverse)), CD248 (5′ TTGCACTGGGCATCGTGTA 3′ (forward), 5′ TTGCTCCCAGCATGGATGAC 3′ (reverse)), CD90 (5′ TGAAAACTGCGGGCTCCGA 3′ (forward), 5′ TGCAAGACTGTTAGCAGGAGAG 3′ (reverse)), and PDPN (5′ CCAGTCACTCCACGGAGAAAG 3′ (forward), 5′ GGCGAGTACCTT CCCGACAT 3′ (reverse)).

### 4.9. Immunoprecipitation of Soluble CD248 from Synovial Fluid

Synovial fluids and sera were collected from rheumatoid arthritis patients during routine joint aspiration. Ethical approval was obtained from the Bro Taf Health Authority prior to commencing the study (LREC 02/4382). All patients were diagnosed as having RA for at least 3 months. Patients used non-steroidal anti-inflammatory drugs (NSAIDs), corticosteroids, and disease-modifying anti-rheumatic drugs (DMARDs, with the exception of TNF inhibitors) either alone or in combination. Fluids were rendered cell-free by centrifugation at 13,000 rpm for 10 min.

To confirm the presence of soluble CD248 in vivo, synovial fluid from an RA patient was pre-incubated three times with 100 mg/mL protein A Sepharose (Sigma) to deplete human immunoglobulins. The sample was then incubated with Prosep G (Millipore, Watford, UK), 3.5 μg of a monoclonal antibody against CD248 (clone P7C5) or CD93 (BIIG-4) as an irrelevant control, and 1% Brij-58 overnight at 4 °C. The immunoprecipitate was washed multiple times in 0.1% Brij-58 and the protein eluted in 50 μL of non-reducing Laemmli buffer. Immunoprecipitated proteins were resolved on a 7.5% SDS-PAGE, electroblotted onto a nitrocellulose membrane, and identified by probing with 0.7 μg/mL of affinity-purified rabbit anti-CD248 followed by an HRPO-conjugated goat anti-rabbit secondary antibody, as described above.

### 4.10. CD248 Expression in Frozen Synovial Tissue by Immunohistochemistry

To identify CD248 expression in synovial MSCs, OA and RA synovial cells were immunostained as described for HFFFs above, with 3.2 μg/mL P1A6 or P7C5 (monoclonal anti-CD248) or affinity-purified rabbit anti-CD248 (3.5 μg/mL), respectively, followed by a FITC-conjugated donkey anti-mouse or TRITC-conjugated donkey anti-rabbit secondary antibody. Staining was visualized under a Leica TCS SP2 or DMLB microscope. To examine CD248 expression in RA synovial tissues, 10 μm sections were fixed in ice-cold acetone for 10 min. Following a brief PBS wash step and a 30 min block in PBS–1% BSA, sections were incubated with either 3.5 μg/mL of affinity-purified rabbit anti-CD248 or monoclonal anti-CD248. Sections were double-stained by incubation with either mouse anti-CD93 (BIIG-4, [2]), mouse anti-CD141 (BD Biosciences), mouse anti-fibroblast (5B5, anti-prolyl 4-hydroxylase, Dako, Carpinteria, CA, USA), mouse anti-smooth muscle cell actin (Serotec, Genève, Switzerland), mouse anti-CD14 (MEM18, V. Horejsi, Prague, Czech Republic), mouse anti-CD31 (1:200, Dako Cytomation Ltd., Glostrup, Denmark), mouse anti-CD90 (Thy-1, Serotec), rabbit anti-CD93 [2], or rabbit anti-VEGF (Chemicon, Hampshire, UK). After two 5 min washes in PBS and a further 30-min incubation in PBS–1% BSA to block non-specific sites, sections were incubated with 5 μg/mL each of TRITC-conjugated donkey anti-rabbit and FITC-conjugated donkey anti-mouse secondary antibodies at room temperature. Slides were washed and mounted with Vectashield and images were viewed under a Leica TCS SP2 or DMLB microscope.

### 4.11. Immunodetection of CD248 in Paraffin Wax Tissue Sections of RA Synovial Tissue

We performed double sequential IHC staining on the Leica Biosystems Bond-III automated staining system, using either the rabbit anti-Hu CD248 (clone E9Z7O, Ref 47948S Cell Signaling technology, Danvers, MA, USA), the rabbit anti-Hu ERG (clone EPR3864, Ventana, Cupertino, CA, USA), or the mouse antiCD45 (mouse monoclonal PA0042, Leica, Wetzlar, Germany). We used a paraffin wax RA tissue section and antigen retrieval was achieved using an EDTA-based solution (ER2, Leica). Revelation was carried out using BOND Polymer Refine Detection kits (Ref Ds9390 and Ds9800) according to the manufacturer’s instructions (Leica, Paris, France).

### 4.12. Statistical Analysis

Statistical analyses were conducted using the computer program GraphPad InStat^®^ version 3.01 for Windows 95/NT (GraphPad Software, San Diego, CA, USA). Experiments were performed in triplicate and mean ± SEM values are provided. *p*-values were determined using an unpaired *t*-test or using two-way ANOVA (Bonferroni test), with *p* < 0.05 considered statistically significant.

## 5. Conclusions

To summarize, we confirmed CD248 was expressed by MSCs in vitro and RA tissue ex vivo. Growth factors and inflammatory cytokines tightly upregulated the release of soluble CD248 in an MMP-dependent manner. Therefore, CD248 may act as a marker of inflammatory MSCs switching from an immunosuppressive to a more aggressive phenotype involved in autoimmune diseases (and cancers). The loss of membrane CD248 by shedding mechanisms may plausibly contribute to this switch.

## Figures and Tables

**Figure 1 ijms-24-09546-f001:**
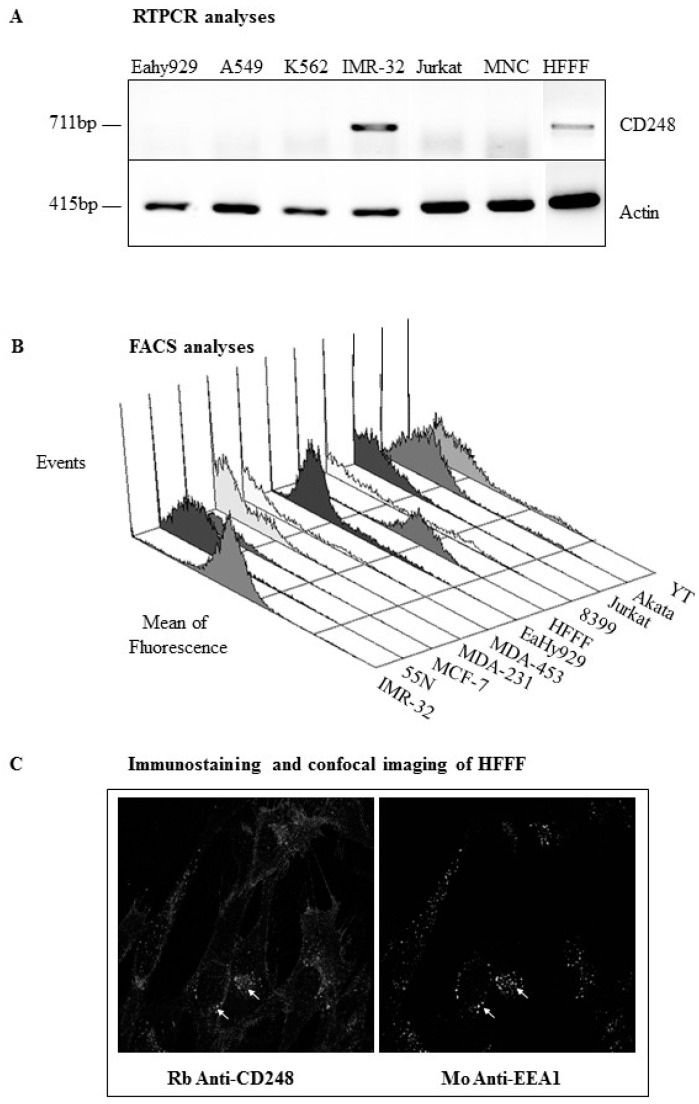
Cell surface and intracellular distribution of CD248 in tumor cells and skin-derived MSCs (HFFFs). (**A**) RNA was extracted from a series of cell lines and CD248 expression examined by RT-PCR using CD248-specific primers. CD248-specific bands (711bp) were observed in the IMR-32 neuroblastoma and HFFF cell lines. Actin was used as a control reaction. (**B**) CD248 expression at the protein level was determined by flow cytometric staining of non-permeabilized cell lines with affinity-purified rabbit anti-CD248 followed by an RPE-conjugated secondary antibody. (**C**) CD248 localization within the cell was determined by double staining of permeabilized (acetone fixed) HFFFs on glass coverslips with affinity-purified rabbit anti-CD248 and a series of anti-organelle markers. TRITC-conjugated donkey anti-rabbit and FITC-conjugated donkey anti-mouse secondary antibodies were used. Expression was localized to the endosomal compartment by double staining with EEA1 and observing colocalization under a confocal microscope (white arrows). Magnification ×630.

**Figure 2 ijms-24-09546-f002:**
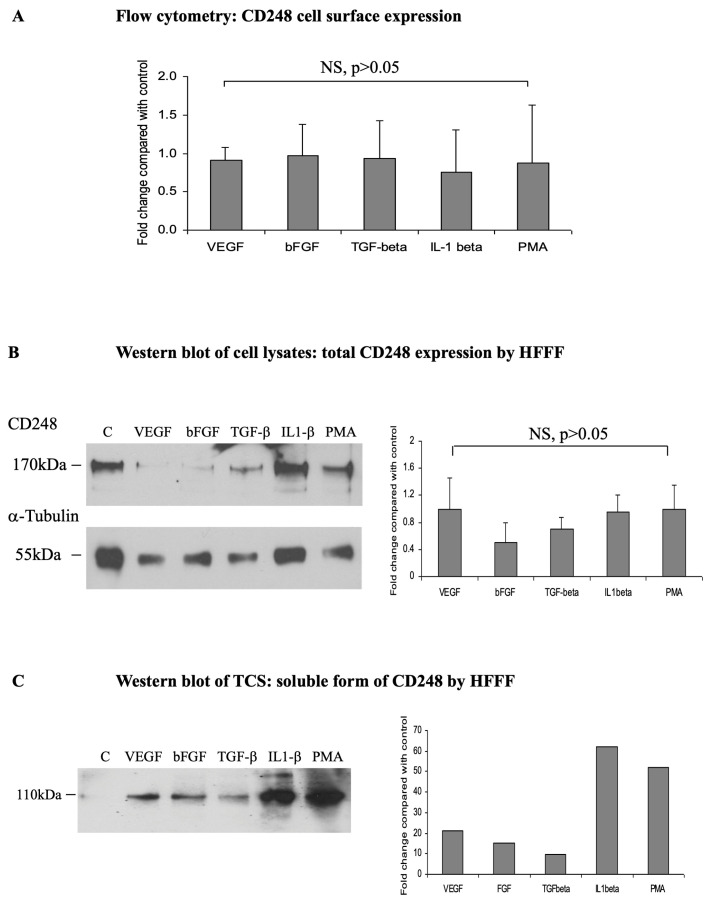
CD248 expression by HFFFs in response to cytokines and growth factors: release of a cleaved soluble form. Quiescent HFFFs were treated for 24 h with 20 ng/mL rhVEGF_165_, 10 ng/mL bFGF, 1 ng/mL TGF-β1, 200 U/mL IL1-β, 50 ng/mL PMA, or were left untreated. (**A**) Flow cytometric staining of cells with affinity-purified rabbit anti-CD248, and expression of the results as fold changes in the mean of fluorescence over non-treated cells. (**B**) Western blot of whole cell lysates with affinity-purified rabbit anti-CD248 and comparison of the intensity of the CD248-specific band (160–170 kDa) following treatment with that of the non-treated cells showed no significant change in total cell CD248 expression following any treatment (*p* > 0.05). Unequal loading was accounted for by blotting for α-tubulin (50 kDa band). The blots are representative of three separate experiments. The graph represents the mean change in intensity compared with non-treated cells ± SEM, *n* = 3. (**C**) TCS was collected from treated cells, freeze-dried and subjected to Western blotting with affinity-purified rabbit anti-CD248. Bands of 110 kDa were observed and the intensities of the bands following treatment were compared to the intensity of the band in non-treated cells. Histograms show the analysis of the band intensity of the WB (*n* = 1). The blot and graph are representative of three separate experiments. The bars represent the fold increase in intensity over non-treated cells.

**Figure 3 ijms-24-09546-f003:**
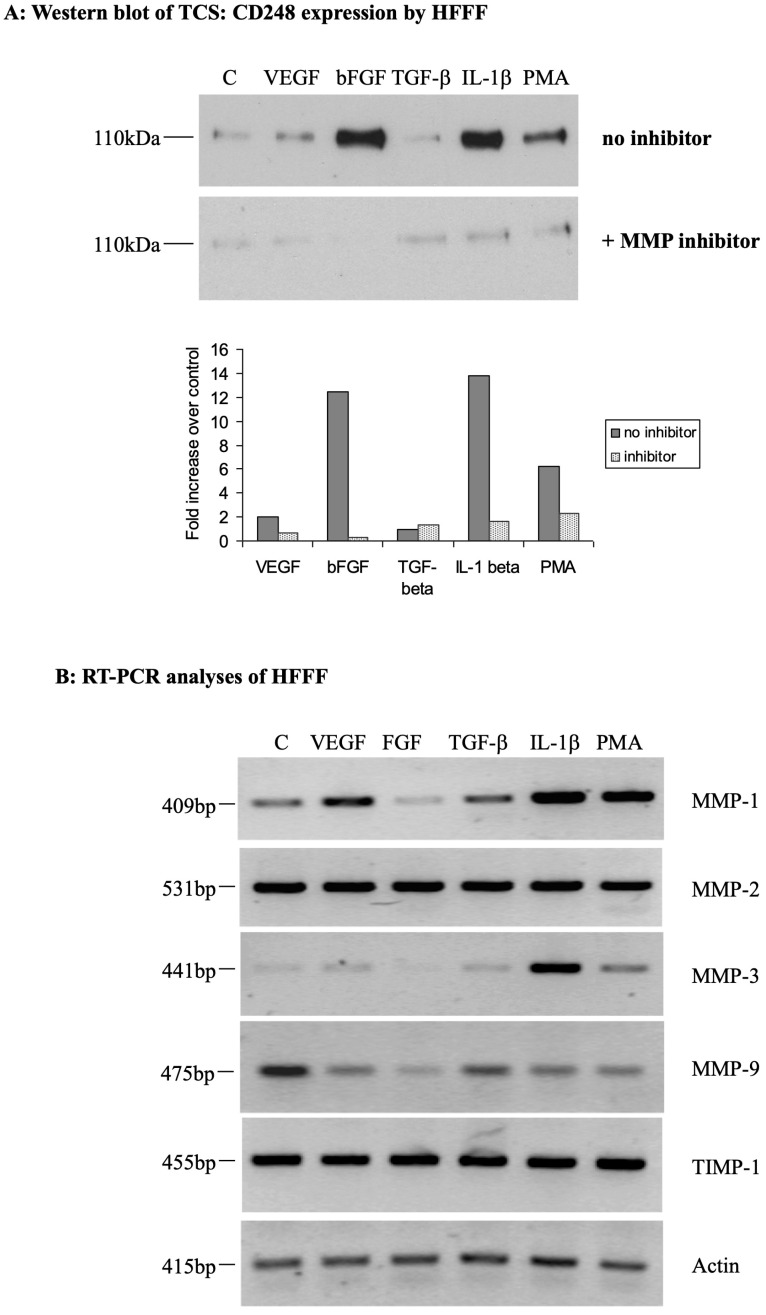
Matrix metalloproteases play a role in the generation of soluble CD248. HFFFs were treated for 4 h with 20 ng/mL rhVEGF_165_, 10 ng/mL bFGF, 1 ng/mL TGF-β1, 200 U/mL IL1-β, 50 ng/mL PMA, or were left untreated. (**A**) Cells were treated for 30 min with 1,10-phenanthroline prior to treatment together with growth factors and cytokines. TCS was collected, freeze-dried, and CD248 expression analyzed by Western blot probing with affinity-purified rabbit anti-CD248. Soluble CD248 is represented as a single band of 110 kDa. Histograms show the analysis of the band intensity of the above WB (*n* = 1). (**B**) RNA was extracted from the cells and subjected to RT-PCR using primers specific for MMP-1, MMP-2, MMP-3, MMP-9, and TIMP-1 as well as actin primers as a control. PCR products were separated on an ethidium-bromide-stained 0.7% agarose gel and visualized under UV light.

**Figure 4 ijms-24-09546-f004:**
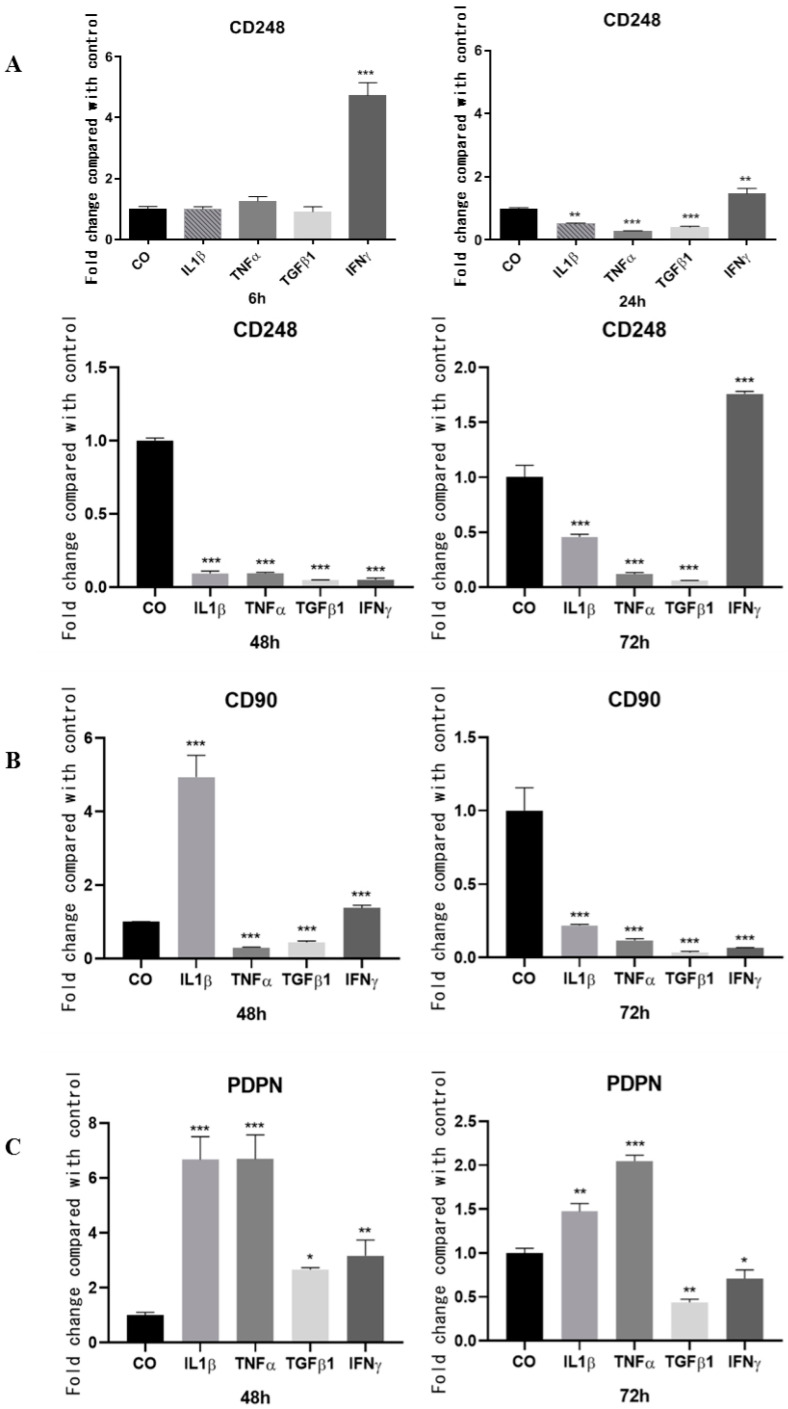
CD248, CD90, and PDPN expression by synovial MSCs in response to inflammatory mediators. MSCs were stimulated with either IL1-β, TNF-α, TGFβ1, or IFN-γ (all at 20 ng/mL) or unstimulated during 6, 24, 48, or 72 h. Total RNA was extracted, and qRT-PCR was performed. Graphs represent the mean change in mRNA expression of (**A**) CD248, (**B**) CD90, and (**C**) PDPN compared to unstimulated MSCs ± SEM, *n* = 3. GAPH was used as the housekeeping gene. * *p* < 0.05, ** *p* < 0.01, and *** *p* < 0.001.

**Figure 5 ijms-24-09546-f005:**
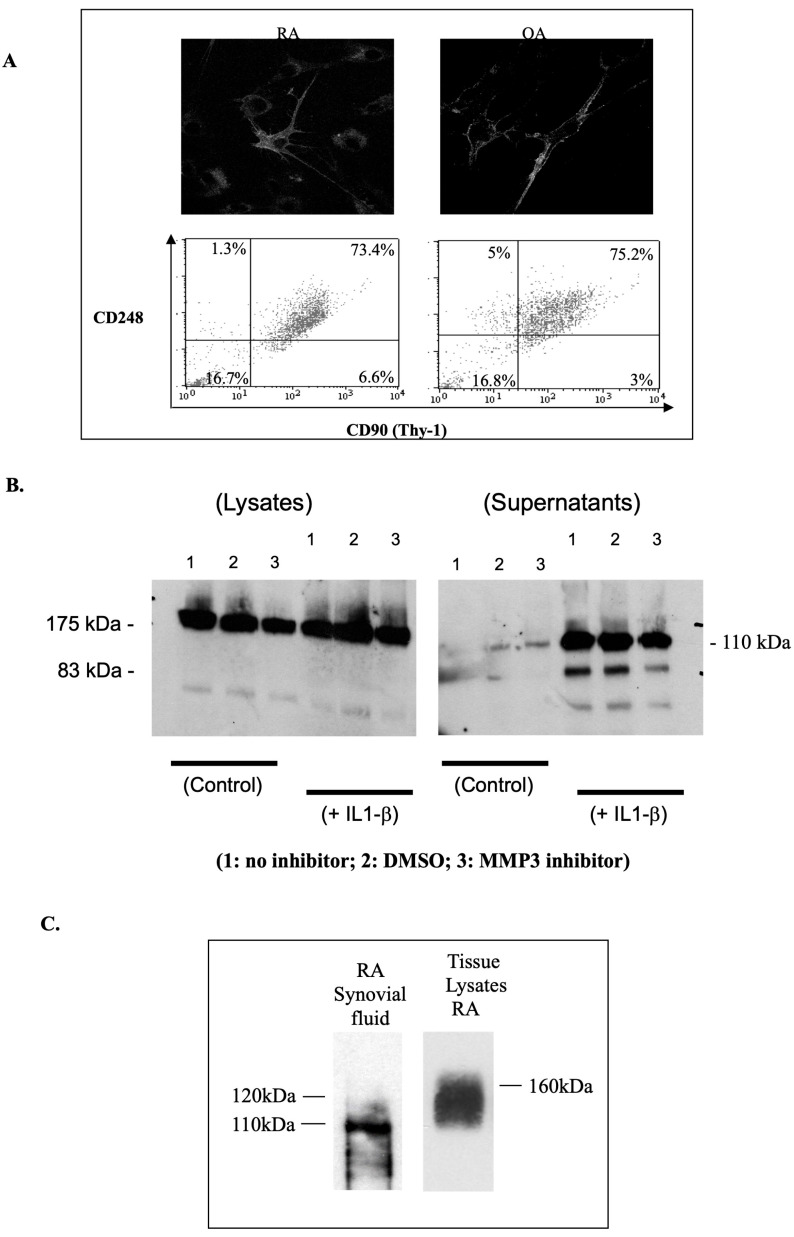
CD248 expression by synovial MSCs from arthritis patients. (**A**) OA and RA synovial MSCs/fibroblasts (RASF or OA SF) were grown on glass coverslips, acetone-fixed, and stained for CD248 using P1A6 (OA) or rabbit anti-CD248 (RA), followed by FITC-conjugated donkey anti-mouse or TRITC-conjugated donkey anti-rabbit respectively. Staining was visualized by confocal microscopy. Magnification ×400. Synovial fibroblasts were double-stained by flow cytometry with affinity-purified rabbit anti-CD248 and mouse anti-CD90 (Thy-1), followed by RPE-conjugated goat anti-rabbit and FITC-conjugated goat anti-mouse secondary antibodies. The staining was analyzed on a FACSCalibur instrument. The percentage of gated cells in each region is indicated. (**B**) soluble CD248 forms (110 kDa major band and two further cleaved products) were also detected in the supernatants of RA fibroblast cultures (passage 4) in response to IL1-β (24 h treatment). However, an MMP-3-specific inhibitor failed to control CD248 cleavage. (**C**) CD248 was immunoprecipitated from the synovial fluid using P7C5 bound to protein G Sepharose. The immunoprecipitated protein was detected by Western blot probing with affinity-purified rabbit anti-CD248, revealing a band of 110 kDa representing soluble CD248 (lane 1). A Western blot of lysates prepared from the synovial tissue of an OA and RA patient revealed a band of >120 kDa.

**Figure 6 ijms-24-09546-f006:**
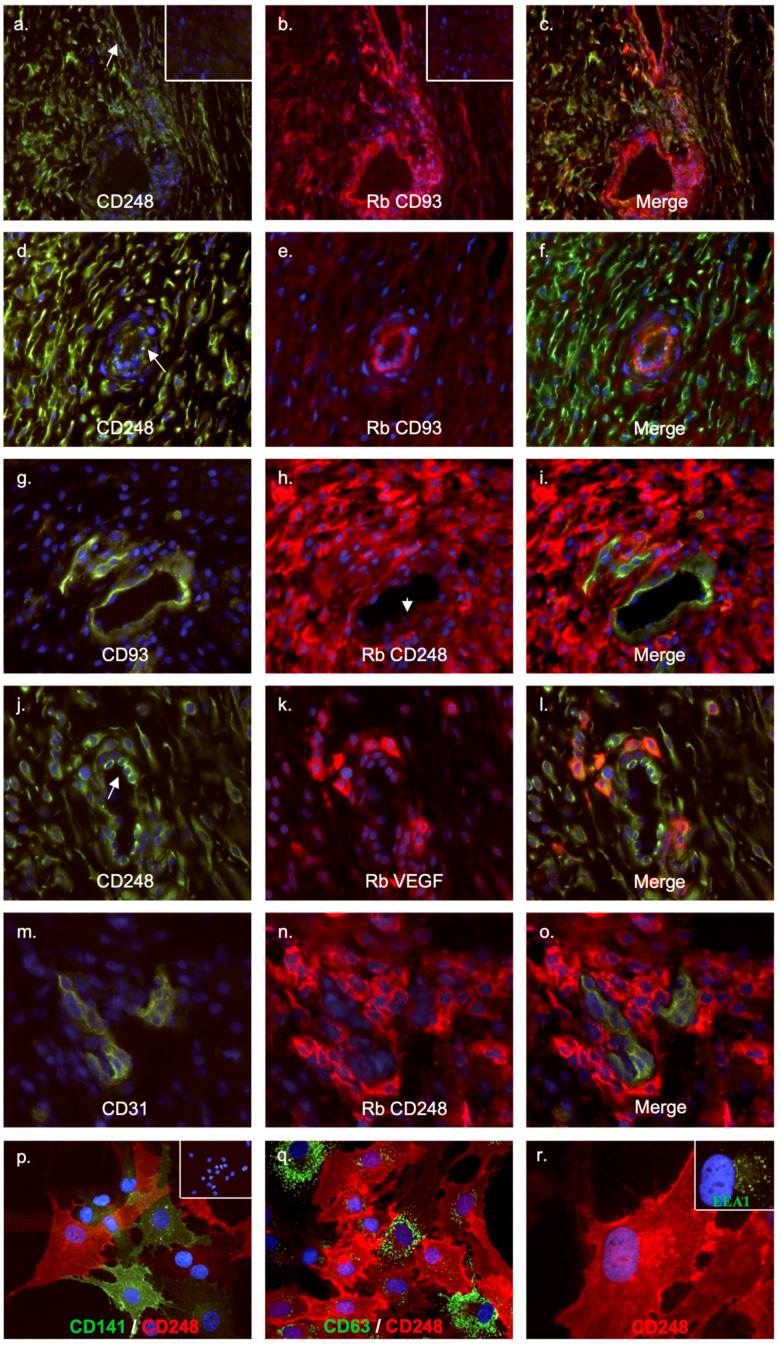
CD248 is abundantly expressed by synovial MSCs in RA inflammatory tissues. (**a**–**o**): Frozen synovial tissue sections from an RA patient were stained with affinity-purified mouse (P1A6) or affinity-purified rabbit anti-CD248 (white arrows). Sections were double-stained using antibodies against CD90 (Thy-1), CD31, CD93, and VEGF followed by TRITC-conjugated and FITC-conjugated. Images were viewed using a fluorescent DMLB microscope. Insets in panels a and b depict background stainings using irrelevant antibodies. Magnification ×100 (**a**–**c**); ×200 (**d**–**f**), and ×400 for panels (**g**–**o**). (**p**–**r**): RA fibroblasts cultured on glass coverslips were double-stained for CD248 (affinity-purified rabbit antibody) and monoclonal antibodies against either CD141, CD63, or EEA1. Magnification (×400 for (**p**,**q**) and ×1000 for panel (**r**)). Irrelevant antibodies failed to stain fibroblast cells (inset, panel (**p**)).

**Figure 7 ijms-24-09546-f007:**
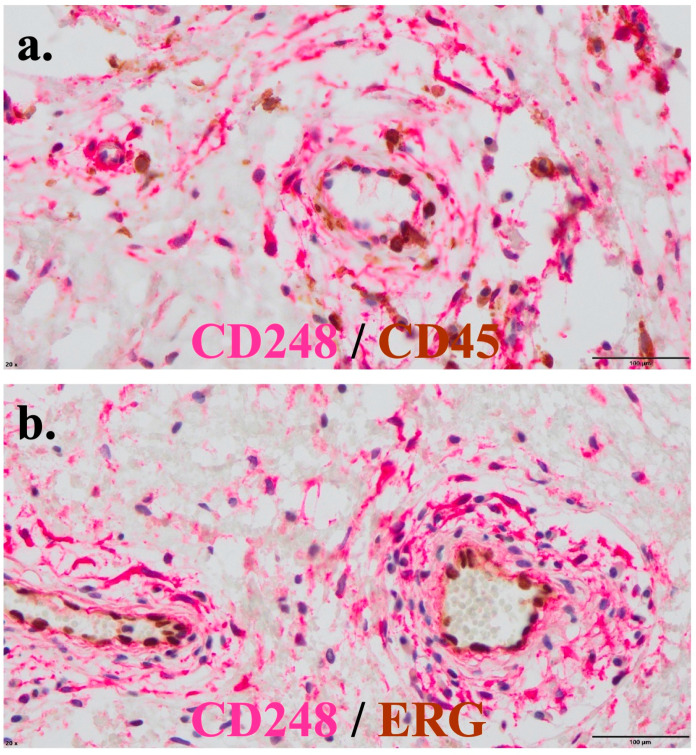
Paraffin wax RA tissue sections double-stained for CD248 and either CD45 (leukocytes) or endothelial (ERG) cell markers. Serial formalin-fixed paraffin-embedded (FFPE) sections were processed using sequential alkaline-phosphatase-based immunostaining with anti-CD248 (clone E9Z7O, red staining) and diaminobenzidine-based immunostaining with (**a**) anti-CD45 (clone X16/99, Leica) or (**b**) anti-ERG (clone EPR3864, Ventana).

**Figure 8 ijms-24-09546-f008:**
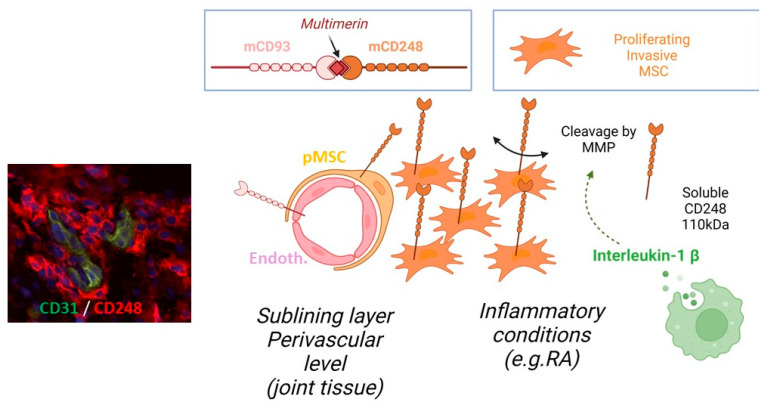
Perivascular synovial MSCs (pMSCs) express CD248 (endosialin) and may remain at the perivascular level through the multimerin–CD93 anchoring complex. In inflammatory conditions (e.g., RA), interleukin-1 beta produced by macrophages and stimulating the release of MMP will contribute to the shedding of CD248. Membrane CD248 is known to control the PDGF-induced proliferation of MSCs (switching to an aggressive phenotype involved in pannus formation) but the role of sCD248 in this process or as a decoy receptor remains to be ascertained. Microscope image is shown at ×400.

## Data Availability

Not applicable.

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
