# Peer review of "Inflammatory Mesenchymal Stem Cells Express Abundant Membrane-Bound and Soluble Forms of C-Type Lectin-like CD248"

_ijms, 2023, doi:10.3390/ijms24119546_

Round 1

Reviewer 1 Report

Brief Overview:

The authors examined the expression of CD248, a glycoprotein, in vitro in skin and synovial mesenchymal stem cells, as well as in tissue samples from rheumatoid arthritis and osteoarthritis patients. The study found that while there was no significant change in membrane expression, a soluble form of CD248 was detected, particularly after treatment with IL1-β and PMA. The release of soluble CD248 was found to be dependent on MMPs. The authors suggest that the shedding of membrane CD248 may act as a marker of inflammatory MSC switching from an immunosuppressive to a more aggressive phenotype.

Major Points:

1.     In Figure 2B, it would be valuable to explain the reason for the unequal loading of the samples. Please measure the protein concentration after cell lysis and load an equal amount of protein in each well. Since there is a drastic change in protein levels between samples, normalizing the CD248 expression to tubulin levels would not result in accurate data. If there was an effect on cell viability following treatments, please report the data in the results section. Additionally, including a 24hr time point would provide further insights. Please consider repeating the western blot in Figure 2B with these suggestions.

2.     Performing a qRT-PCR for CD248 in HFFF cells following treatment with rhVEGF165, bFGF, TGF-b1, IL-1b, and PMA at 1.5 hr, 4 hr, and 24hr would be valuable. As the authors observe that IL-1b and PMA increase soluble CD248 and maintain membrane-bound levels of CD248, mRNA levels could provide valuable insights into CD248 regulation.

3.     In section 2.4, it would be valuable to determine whether CD248 mRNA expression is different at early time points (4hr and 24hr). Please explain why a 20 ng/ml concentration was selected for all cytokines. Did the authors perform titrations to determine the above-mentioned concentration? In Figure 2, 200U/ml IL-1b was used. Please explain why?

4.     In Figure 5B, it would be valuable to include an MMP-1 specific inhibitor, as IL-1B also upregulates MMP-1 as shown in Figure 3B. Additionally, including a broad spectrum MMP inhibitor would provide further insights.

Minor Points:

1.     For Figure 2, please include all the data from the 24-hr treatment as a part of the supplemental information. Any data indicated as “data not shown” in the article can be included in the supplemental information.

2.     For Figure 2C and 3A, please add error bars. Please explain why the soluble CD248 fold increase over control is much higher in Figure 2C when compared to 3A with no inhibitor.

3.     In the Figure 3B title, please change "RTCPR" to "RT-PCR."

4.     Regarding Figure 5C, please explain why the CD248 band from the lysates is lower in molecular weight than the CD248 band in Figures 2B and 5B. This explanation can be included in the discussion section.

5.     In lines 346-347, please avoid including unpublished observations without data in the manuscript. Instead, authors can include future directions of this study in the discussion section.

Reviewer 2 Report

Reviewer comments

This study elucidated the mechanism of CD248 expression in vitro using various treatment of growth factors and cytokines, MMP inhibitors, and identified a soluble form of CD248 (sCD248) was formed when cells were treated with IL1-beta and PMA under the inflammatory condition. This study articulated the plausible mechanistic insights for inflammatory factors induced shedding of membrane bound CD248 of mesenchymal stem cells and shows great potentials to guide further mechanistic investigations. The contents fit the scope of the journal and the manuscript was well-organized.

I have a few comments, both major and minor, to further improve the quality of the manuscript.

Major:

1.     Figure 2C no SEM and statistical test shown on the bar plot. In the legend, it says “The blot and graph are representative of three separate experiments.” Please add in SEM error bars and statistical test as shown in Figure 2A and 2B bar plots.

2.     Figure 3A, same comments as above, please add in SEM and statistical test on the graph.

3.     Figure 2C and Figure 3A, it seems the TCS was used for the respective analysis. If possible, should include certain internal standard (although might be challenging to find one). The bFGF induced CD248 expression effect without MMP inhibitor looks much stronger in Figure 3A, as compared to Figure 2C. Might consider spike some internal references? If not, please discuss as a limitation.  

4.     Section 5.6, please provide more details on the rationale behind the dose selected for respective growth factor, cytokines and MMP inhibitors.

Minor:

1.     It would be much clearer to summarize the primer used for RT-PCR as a table.

2.     In abstract, “A soluble (s) form of cleaved CD248…” not sure why there is a “(s)” there.

3.     Page 2, “The inflammatory MSC contribute to the pathology of RA by releasing soluble mediators, including cytokines and proteolytic enzymes (including matrix metalloproteases (MMPs), cathepsins, and serine proteases, which directly or indirectly promote cartilage degradation [21,22].”. Lack a right parathesis somewhere.

4.     Figure 2 legend, lower case “…Release of a 134 cleaved soluble form.”

5.     Page 5 “we incubated the cells for 30 151 minutes prior to treatment with the growth factors with a broad spectrum MMP inhibitor 152 1,10-phenanthroline [24].” There shouldn’t have reference in the results section?

6.     “To determine further the expression and the regulation of CD248 together with that 178 of canonical synovial MSC markers ie CD90 and podoplanin (PDPN)”

“ie CD90 and podoplanin (PDPN)” should have parathesis like “(i.e., CD90 and podoplanin (PDPN))”?

7.     Section 2.5, abbreviation “FLS” first appearance should have full name.

8.     A notation should be added for sCD248 when it first appeared for “soluble CD248”

9.     Page 14, reference formatting “{Wang, 2002 #41}”. Please re-check the manuscript formatting throughout the manuscript.

See above comments under "Minor"

Round 2

Reviewer 1 Report

Thank you very much for addressing the Comments.

Minor Points:

1.   In lines 179-182, please state that CD248 mRNA levels were also measured at 6h and 24 hours.

2.      In line 182, please revise the sentence to state that the treatments did not show any cytotoxic effects on the cells.

3.      For Figure 2C and 3A, please add error bars. Please explain why the soluble CD248 fold increase over control is much higher in Figure 2C when compared to 3A with no inhibitor.

Reviewer 2 Report

Reviewer comments

Thanks for addressing my previous comments. I have a few additional comments.

1.     If no SEM could be added to Figure 2C and Figure 3A. The author should either removing these figures or discuss this as a limitation (unable to perform relevant experiments involving HFFF, etc)

2.     Resolution of Figure 3 looks really poor following the revision. Need to upload with a higher resolution pic.

3.     Figure 4 should be adjusted more consistently, some of the figures seems go out of the margin. Some have borderlines, others don’t.

4.     Page 17, type in “2?g/mL of the monoclonal anti- 438 CD248 antibody (clone P7C5)…”

See above
